# Soil acidity and fertility status of surface soils under different land uses in Sayo district of Oromia, western Ethiopia

**Abu Regasa**[1,2]*, **Wassie Haile**[2], **Girma Abera**[3]

**1** Department of Natural Resource Management, College of Agriculture and Natural Resource, Dambi Dollo University, Dambi Dollo, Ethiopia, **2** School of Horticultural Plant Science, College of Agriculture, Hawassa University, Hawassa Ethiopia, **3** Ethiopian Agricultural Transformation Institute (ATI), Addis Ababa, Ethiopia

* aburegasa16@gmail.com

**Data Availability Statement:** All relevant data are within the manuscript and its Supporting information files.

**Funding:** The author(s) received no specific funding for this work.

## Abstract

Land use conversion from natural forests to grassland, plantation forests, mono-cropping coffee and croplands is a significant causes of soil degradation, leading to aggravate soil acidity and nutrient depletion. However, there is limited information regarding comprehensive effect of land use conversion on soil fertility and acidity in western Oromia Region of Ethiopia. Hence, this study aims to assess the surface soil fertility and acidity across different land use types (forest, crop, eucalyptus land, grazing land, and coffee farmland) to provide management options. A total of 60 composite soil sample were collected from four villages representing these land uses and analyzed for selected soil fertility and acidity indicators. Accordingly, sand content was highest in eucalyptus lands, whereas clay content high in forestlands. The highest soil bulk density and exchangeable acidity were observed in eucalyptus lands. The lowest pH was observed in cropland soils and the highest in forestland soils. Organic matter, total nitrogen, and available phosphorus were low in eucalyptus lands and high in forestland soils. Forest and coffee farm lands showed higher exchangeable cations, cation exchange capacity, and percentage base saturation, whereas cultivated, grazing, and eucalyptus lands obtained high level of micronutrients and exchangeable acidity. Generally, the findings indicate that land conversion has caused for lower soil pH and diminished essential macronutrients. These negative impacts on soil quality emphasize the need for sustainable soil management practices to mitigate soil degradation includes soil acidity and fertility, which significantly improve agricultural production and environmental health.

## Introduction

Ethiopia is implementing an agriculture-based development strategy aimed at addressing the pressing challenges of food insecurity, widespread poverty, and the need for national economic growth. Agriculture has historically been, and continues to be, a cornerstone of the Ethiopian economy, accounting for 32.4% of the country's GDP, 80% of its total

**Competing interests:** The authors have declared that no competing interests exist.

exports, and employing an estimated 75% of the labor force, mostly in rural areas [1]. Given its critical role, the agricultural sector is seen as a key driver of economic transformation, with the potential to lift millions out of poverty, improve livelihoods, and ensure food security for a growing population [2, 3]. In essence, enhancing agricultural productivity and reducing the vulnerability of this sector to external shocks such as climate change, erratic rainfall, and market fluctuations are viewed as essential for the country's development [4]. To boost agricultural production, the Ethiopian government has undertaken several initiatives aimed at modernizing small-scale farming, which dominates the agricultural landscape. These efforts include the distribution of fertilizers and improved seeds to farmers, targeting crops that are more resilient and higher yielding, as well as training programs on best practices for soil management, pest control, and climate-smart agricultural techniques [5]. Additionally, the government has been promoting irrigation projects to reduce dependency on rainfall and mitigate the effects of droughts, which are becoming more frequent due to climate change. These interventions have contributed to a gradual increase in productivity, and Ethiopia has set an ambitious target of increasing annual crop production by 60% between 2021 and 2030, which, if successful, would significantly strengthen food security and export potential [6].

Despite these efforts, Ethiopia faces major obstacles to achieving its agricultural development goals, with land degradation being one of the most significant challenges. The degradation of land and soil is widespread, severely undermining agricultural productivity and threatening the sustainability of rural livelihoods [7–9]. The country's landscapes are marked by various forms of degradation, including severe soil erosion, nutrient depletion, loss of organic matter, increased soil acidity, and salinity, all of which contribute to reduced soil fertility [10]. In addition to these soil-specific issues, broader environmental challenges such as deforestation, landslides, and the degradation of wetlands and water bodies are further aggravating the situation, leading to a vicious cycle of ecosystem degradation [11]. A major contributor to this degradation is unsustainable land-use practices, driven by a growing population and the increasing demand for agricultural land. Large areas of Ethiopia's natural forest cover have been cleared and converted into grazing areas, eucalyptus plantations, and cropland, resulting in significant changes to soil properties [12]. The conversion of land from its natural state, particularly in the Ethiopian highlands, has led to the deterioration of soil structure, reduced water retention, and the loss of important nutrients [13]. This shift in land use, coupled with poor land management practices, has not only diminished soil productivity but also increased vulnerability to climate-related shocks, such as floods and droughts, further threatening the long-term sustainability of the agricultural sector.

Soil fertility depletion is a significant barrier to sustainable agriculture and is a key factor contributing to the slow growth in food production across Sub-Saharan Africa (SSA), including Ethiopia [4]. Ethiopia is particularly vulnerable to soil fertility loss due to its mountainous terrain and reliance on small-scale, intensive cereal farming systems [14]. Several independent studies have confirmed that the country faces a negative soil nutrient balance, with alarming rates of nutrient depletion. On average, Ethiopian soils lose 122 kg of nitrogen (N), 13 kg of phosphorus (P), and 82 kg of potassium (K) per hectare annually, primarily due to soil erosion and nutrient uptake by crops [15]. This ongoing nutrient loss severely compromises the fertility of the soil, further threatening agricultural productivity and food security in the region. Soil acidity is the other major agricultural challenge in Ethiopia, affecting nearly half of the country's arable land and significantly reducing the productivity of important crops. Around 43% land of Ethiopia affected by soil acidity, 28% is considered moderately to weakly acidic (with a pH range of 5.5 to 6.7), while 15% is strongly to moderately acidic, with a pH of less than 5.5 [16, 17]. These strongly acidic soils are particularly problematic, as they create conditions

where essential nutrients like phosphorus, calcium, and magnesium become less available to plants. Moreover, approximately one-third of these acidic soils suffer from aluminum toxicity, which further restricts plant growth by damaging root systems and limiting nutrient uptake [18]. In regions like Assosa and Wolega, the soil acidity issue is even more pronounced, with soils ranging from moderately to very strongly acidic. These acidity is largely attributed to the high levels of rainfall in western Ethiopia, which results in substantial leaching of essential nutrients, particularly in areas where the soils are already inherently poor in buffering capacity. Rainfall in these regions exceeds 1000 mm per year, accelerating the acidification process by washing away basic cations such as calcium and magnesium, leaving behind an acidic soil environment. Additionally, the limited availability of soil management technologies such as liming, the use of appropriate fertilizers, and access to acid-tolerant crop varieties exacerbates the problem, making it difficult for farmers to mitigate the impact of acidity on their crop yields [4]. The widespread presence of acidic soils presents a significant challenge to agriculture, requiring interventions such as liming and the use of acid-tolerant crop varieties to improve productivity.

Soil fertility depletion and increased acidity are often driven by land management practices, particularly the conversion of land from its natural state. The variation in soil acidity and fertility depletion across different land use types highlights how land use changes can aggravate soil degradation, especially at the soil surface. Several studies have shown that converted lands such as those used for cultivation, eucalyptus plantations, and grazing tend to have lower soil pH, reduced base saturation, higher exchangeable acidity, and altered micronutrient levels compared to undisturbed, natural ecosystems like forests [19–21]. These findings underscore that land conversion worsens soil acidity and fertility loss, primarily due to factors such as increased nutrient leaching, removal of organic matter, and disruption of the natural soil balance. In cultivated soils, for instance, intensive farming and the lack of replenishment of essential nutrients lead to significant declines in soil health. Similarly, the expansion of eucalyptus plantations, known for their high water and nutrient uptake, further depletes soil fertility and accelerates acidification [21]. Therefore, understanding the relative distribution of surface soil acidity and fertility across various land use patterns is essential for developing sustainable soil management strategies. Through assessing how different land uses influence soil properties, land managers can implement targeted practices to restore soil health, such as incorporating organic amendments, rotating crops, or reintroducing vegetation cover. These methods are fundamental for ensuring long-term soil productivity and sustainability, particularly in regions vulnerable to soil degradation [10].

In the Oromia Region, particularly in the western highlands, there has been limited research on surface soil acidity and fertility under various land uses [22]. This is especially true for the Sayo district, where no comprehensive studies have assessed the surface soil acidity and fertility status. Since the early 1980s, natural forestlands in the area have been gradually converted into coffee farms, croplands, grazing lands, and eucalyptus plantation forests. However, the effects of these land use changes on soil acidity and fertility remain scarcely understood, creating a significant knowledge gap. This lack of information hinders effective soil management, potentially exacerbating soil acidity and fertility decline across the district. Without precise data, the ongoing processes of soil acidification and nutrient depletion risk being overlooked, threatening to spread and degrade additional areas. To address these challenges, this study aimed to evaluate the status of surface soil acidity and fertility under different land use types in the Sayo district of western Oromia Region, Ethiopia. Through conducting a site-specific assessment, the research seeks to provide critical insights into the district's current soil conditions, supporting efforts to mitigate soil acidity and prevent further degradation.

## Material and methods

### Description of the study area

This study was conducted in Kellem Wollega zone of Oromia Region in western Ethiopia. Geographically, it is located between 8˚10'00" to 9˚20'00″ N latitude and 34˚10'00" to 35˚ 20'00″ E longitude (Fig 1). The seasonal and spatial variations in temperature depend on altitude gradients. Accordingly, the lowest and highest monthly average temperatures were 16 and 28˚C, respectively. The area receives an average yearly rainfall of 1610 mm. The rainfall pattern was unimodal and extended from April to October (Fig 2). The study site has an undulating, rugged, and hilly topography with altitudes ranging from 530 to 2346 m [23]. According to the FAO digital soil map of the world, the major soil types in research site are Nitisol and Cambisol, which prevail in the northern and southern study site, respectively [24]. The land-use patterns of the study site are diverse, including crop fields, plantation forest, grazing lands, forestland, coffee plantations, and other infrastructure. A significant portion of the study site was enthusiastic about crops and coffee farming. Additionally, the area has a mixed farming system that incorporates both crop cultivation and livestock rearing. The area is well suited for growing a diverse range of crops, including maize *(Z. mays)*, sorghum *(Sorghum bicolour)*, wheat (*Triticum aestivum*), haricot bean (*Phaseolus vulgaris*), oat (*Avena sativa*), peas (*Pisum sativum*), and fava bean (*Vicia faba*). In addition, ensete (*Ensete ventricosum*), coffee (*Coffea spp.*), and khat (*Catha edulis*) are growing in the study area. Coffee, in particular, is a major cash crop that is extensively cultivated, especially in areas near or under forested lands [23].

### Site selection and soil sampling

The study sites and land use types (LUTs) were selected following a reconnaissance survey and field observation conducted in 2022/23 during the off-season, with guidance from local elders and development agents of the district. Through this survey, the district's overall geography was assessed. Following this, five dominant land use/cover types were recognized, including

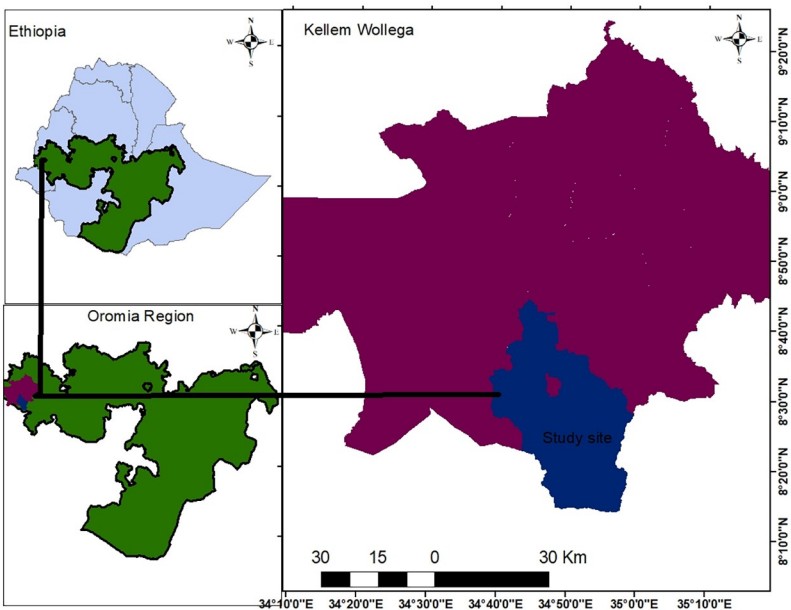

**Fig 1. Map of the study area (source: https://earthexplorer.usgs.gov/).**

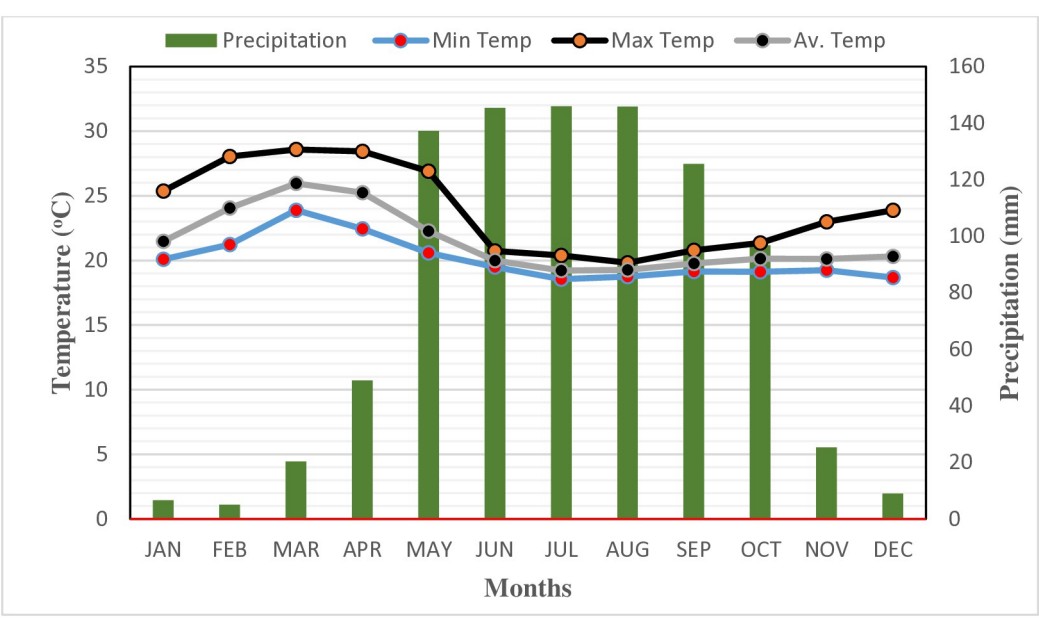

**Fig 2. Mean monthly temperature and precipitation of the study area recorded from 2000–2022 GC.** Source: National Meteorological Service Agency from (S1 Table).

natural forest land (land covered by forests that have developed naturally without significant human intervention), coffee farmland (areas of land specifically cultivated for growing coffee plants), grazing land (land primarily used to support the grazing of livestock such as cattle, sheep, goats, and other domesticated animals), cultivated land (land that has been prepared and managed for the purpose of growing annual crops) and eucalyptus plantation land (refers to areas of land dominated by Eucalyptus trees, cultivated for specific purposes). These land use types were selected from four villages namely Abba Jarra, Alaku Dorgomme, Abichu Shogo, and Tabor based on their similar agro-ecological conditions, the presence of various adjacent LUTs, and their representativeness for the study.

Soil sampling was conducted during dry season in Ethiopia, specifically in January 2022, for several reasons. First, collecting soil in dry conditions makes digging and handling easier, reduces the chance of errors caused by soil compaction or settlement, and ensures consistent sampling conditions, allowing for more reliable comparisons. It also minimizes the risk of contamination due to lower microbial activity and helps preserve soil structure during transport and storage [25]. Additionally, this period is ideal as most crops are harvested by January, leaving the fields clear. Traversing the area and collecting samples is also more manageable during the dry season, avoiding the challenges posed by muddy and sticky conditions. The required soil samples were collected from the identified land-use types.

Soil samples were collected from the dominant land use types in the area by gathering composite samples from the surface soil (0–20 cm) within a 30m × 30m square plot. An auger was used to obtain samples from the four corners and the centre of each plot. The decision to use a consistent sampling depth of 0–20 cm across all land use types was based on several key factors. Firstly, the top 20 cm of soil is widely regarded as critical for soil studies due to its high nutrient content and vulnerability to the effects of land use changes [25]. Secondly, maintaining a standardized sampling depth ensures that all samples are collected under similar conditions, enabling reliable comparisons of how different land uses affect soil properties. Thirdly, using a uniform depth increases measurement accuracy and reduces variability between

samples. This topsoil layer is also the most biologically active, making it essential for study. Additionally, since the Ethiopian traditional plough typically reaches a depth of 20 cm, sampling at this depth allows for meaningful comparisons across different land use types. Finally, using a consistent sampling depth simplifies the process and ensures methodological standardization [26]. In addition, undisturbed core samples from each sampled land use type were taken at the centre of the selected plot with a cylindrical metal core sampler (volume = 100 cm$^3$). All sampling points were georeferenced by the Global Positioning System (GPS).

## Soil analysis

The collected soil samples were placed in a plastic bag, labelled, sealed, and transported to the soil laboratory for preparation and analysis. Then, the soil samples were air-dried at room temperature, ground with a mortar and pestle, and sieved with 2 mm for soil parameters determination except soil OC and total N, which then passed through a 0.5 mm screen. Accordingly, soil particle size distribution was identified by the hydrometer method [27]. The soil bulk density was determined from undisturbed soil samples following the core sampling method whereas, particle density was determined by the pycnometer method [25]. Finally, soil total porosity (TP) was calculated from the average values of bulk and particle density as follows equation:

$$TP(\%) = \left(1 - \left(\frac{BD}{PD}\right)\right) * 100 \tag{1}$$

Where; BD is bulk density, PD is particle density & TP is Total Porosity.

The soil pH was determined with water ($H_2O$) solution using a 1:2.5 soil-to-water ratio [25] whereas the organic carbon was determined by wet digestion method [28]. The soil organic matter content was calculated from organic carbon (Eq 3). The total nitrogen content of the soil was determined by the wet-oxidation procedure of the Kjeldahl method [29] whereas available P was extracted by the Bray-II method using 0.03M of $NH_4F$ and 0.1M of HCl solution [30]. The carbon to nitrogen ratio was computed from the soil organic carbon and total nitrogen (Eq 4). An Atomic absorption spectrometry was employed to measure exchangeable calcium and magnesium, while a flame photometer was employed to measure potassium and sodium [25]. Cation exchange capacity was determined by ammonium acetate method at pH 7.0 [31]. Base saturation was calculated using the total of exchangeable basic cations (Eq 5) and cation exchangeable capacity (Eq 6). Total exchangeable acidity was determined by saturating the soil samples with 1 M KCl solution and from the same extract, exchangeable Al in the soil samples was determined by the application of 1 M NaF [25] whereas acid saturation content was computed from exchangeable acidity and ECEC (Eq 2). The extractable micronutrients (Fe, Zn, Mn, and Cu) were extracted from soil samples using DTPA [32]. Atomic absorption spectrometry was employed to measure the concentrations of all extracted micronutrients (Fe, Zn, Mn, and Cu).

The following parameters were computed from the result of the chemical analysis:

$$AS\,(\%) = \frac{Exchangeable\ acidity}{ECEC} * 100 \tag{2}$$

$$OM(\%) = OC * 1.724 \tag{3}$$

$$C:N\ ratio = OC(\%)/TN(\%) \tag{4}$$

$$\text{The sum of Ex. Bases} = \text{Ex. Ca} + \text{Ex. Mg} + \text{Ex. K} + \text{Ex Na} \tag{5}$$

$$\text{PBS} = \frac{\text{Sum of Ex. Bases}}{\text{CEC}} * 100 \tag{6}$$

Where OM = organic matter, OC = organic carbon, TN = total nitrogen, C: N = carbon to nitrogen ratio, PBS = the percentage of base saturation, CEC = cation exchangeable capacity, ECEC = Effective cation exchangeable capacity.

## Statistical analysis

The analysis of variance was employed to test differences in soil physicochemical properties across land-use types after collecting and organizing available soil data. With the help of SAS software version 9.2, the Least Significant Difference (LSD) was employed to separate means for statistically different soil parameters with a probability of 5% (p≤0.05) among land use types. Moreover, simple correlation analysis was executed by calculating Pearson's correlation coefficient to evaluate the degree and magnitude of associations between various soil acidity indicators and soil physiochemical properties [33].

## Ethics statement

Dambi Dollo University's Research and Publication Directorate and Hawassa University's Research and Publication Directorate authorized the present study to collect soil samples and access the field site. Farmers agreed to collect the soil samples in the study area, as the survey has no harmful effects on humans.

## Results and discussion

### Effects of land use types on soil physical properties

The analyzed soil result illustrated that the soil texture was significantly (p<0.05) varied among land use types (LUTs) (Table 1). Accordingly, the sand proportion of the investigated soil ranged from 24% in the soil of natural forest land (NFL) to 59% in the soil of eucalyptus plantation land (EuPL). The significantly highest sand content was found in the soil of EuPL followed by grazing land (GzL) of the research area. The silt percentage of the studied soil ranged from 10% found in GzL to 24% in soil under cultivated (CuL) and EuPL. The clay fraction ranged from 24% recorded in the soil under EuPL to 60% found in the soil of NFL. Comparing the soil texture under different land uses, numerically higher sand contents of the soil were observed under CuL, GzL and EuPL. Conversely, the soil under these land uses obtains a lower clay fraction than the soil under NFL and CoFL.

The higher sand and lower clay fraction under the soil of the EuPL, CuL and GzL might be due to the loss of fine particles by high rainfall of the area, destruction of soil aggregate with continuous cultivation and animal trafficking, erosion and comparatively low soil organic matter that led to dispersion of soil particles. Conversely, the high clay fraction in the soil under NFL and CoFL might attributed to vegetation cover that saves from erosion and high organic matter content from plant residues that improve soil structure. This shows the strong relationship between clay and soil organic matter which was confirmed by a highly significant positive correlation (r = 0.92**) (Table 4). In agreement with the present finding, [34] found higher clay fraction and lower sand content under the soil of forestland than the soil of cultivated, grazing and shrub lands of north-western Ethiopia. In addition [35, 36] reported higher clay fraction in the soil of forest land than in soil of grazing and cultivated lands. The higher

**Table 1. Effects of land use changes on selected soil physical properties of different villages in Sayo district.**

| LUT | Sand | Silt | Clay | Si/C | Textural class | PD | BD | TP | Sand | Silt | Clay | Si/C | Textural class | PD | BD | TP (%) |
|---|---|---|---|---|---|---|---|---|---|---|---|---|---|---|---|---|
| | % | | | | | g/cm³ | | | % | | | | | g/cm³ | | |
| | | | | | **Aba Jarra** | | | | | | | | **Alaku Dorgomme** | | | |
| NFL | 30[b] | 19[ab] | 51[a] | 0.36[c] | C | 2.4[c] | 0.9[d] | 62[a] | 24[d] | 19[b] | 57[a] | 0.33[d] | C | 2.35[c] | 0.9[c] | 62[a] |
| GzL | 48[a] | 18[b] | 34[b] | 0.54[b] | SC | 2.58[a] | 1.35[b] | 47[c] | 52[a] | 19[b] | 29[c] | 0.65[b] | SCL | 2.53[a] | 1.47[a] | 42[d] |
| CuL | 53[a] | 16[b] | 31[b] | 0.51[b] | SCL | 2.48[b] | 1.36[b] | 45[d] | 46[b] | 24[a] | 30[c] | 0.8[a] | SCL | 2.46[a] | 1.27[b] | 48[c] |
| CoFL | 37[b] | 16[b] | 47[a] | 0.4[c] | C | 2.3[d] | 0.95[c] | 58[b] | 31[c] | 23[a] | 46[b] | 0.5[c] | C | 2.3[c] | 0.94[c] | 58[b] |
| EuPL | 49[a] | 23[a] | 28[b] | 0.8[a] | SCL | 2.58[a] | 1.41[a] | 45[d] | 48[ab] | 24[a] | 28[c] | 0.86[a] | SCL | 2.52[a] | 1.47[a] | 42[d] |
| **Mean** | **43** | **19** | **38** | **0.51** | - | **2.47** | **1.20** | **52** | **40** | **22** | **38** | **0.63** | - | **2.43** | **1.22** | **50** |
| **LSD (0.05)** | **7.84** | **4.2** | **5** | **0.12** | - | **0.05** | **0.04** | **1.58** | **5.5** | **2.6** | **4.3** | **0.09** | - | **0.05** | **0.07** | **2.5** |
| **F** | **14.78** | **4.23** | **39.86** | **22.04** | - | **41.68** | **340.16** | **266.02** | **46.77** | **9.95** | **90.73** | **52.79** | - | **42.43** | **129.41** | **123.19** |
| **P value** | **0.00** | **0.03** | **.0001** | **.0001** | - | **.0001** | **.0001** | **.0001** | **.0001** | **0.00** | **.0001** | **.0001** | - | **.0001** | **.0001** | **.0001** |
| **CV (%)** | **9.9** | **12.5** | **7.2** | **13** | - | **1.3** | **1.9** | **1.7** | **7.5** | **6.8** | **6.2** | **8.2** | - | **1.1** | **3.4** | **2.8** |
| | | | | | **Abichu Shogo** | | | | | | | | **Tabor** | | | |
| NFL | 24[d] | 16[a] | 60[a] | 0.3[c] | C | 2.3[c] | 0.9[d] | 61[a] | 32[d] | 18[a] | 50[a] | 0.37[a] | C | 2.46[c] | 0.95[c] | 61[a] |
| GzL | 56[a] | 17[a] | 27[d] | 0.6[ab] | SCL | 2.5[a] | 1.27[b] | 50[c] | 54[ab] | 10[c] | 37[c] | 0.27[b] | SC | 2.6[a] | 1.15[b] | 56[b] |
| CuL | 51[b] | 14[a] | 35[c] | 0.4[bc] | SC | 2.4[b] | 1.25[b] | 49[c] | 49[b] | 13[b] | 38[c] | 0.3[ab] | SC | 2.5[abc] | 1.22[ab] | 52[c] |
| CoFL | 36[c] | 16[a] | 48[b] | 0.3[c] | C | 2.4[b] | 1.04[c] | 57[b] | 41[c] | 14[b] | 45[b] | 0.3[ab] | C | 2.47[bc] | 0.97[c] | 61[a] |
| EuPL | 59[a] | 17[a] | 24[d] | 0.7[a] | SCL | 2.5[a] | 1.45[a] | 42[d] | 57[a] | 12[bc] | 31[d] | 0.38[a] | SCL | 2.55[ab] | 1.28[a] | 50[d] |
| **Mean** | **45** | **16** | **39** | **0.46** | - | **2.44** | **1.18** | **52** | **47** | **13** | **40** | **0.34** | - | **2.52** | **1.11** | **56** |
| **LSD (0.05)** | **4.8** | **NS** | **4.5** | **0.2** | - | **0.06** | **0.09** | **2.5** | **4.9** | **3.1** | **4.7** | **0.1** | - | **0.07** | **0.06** | **1.8** |
| **F** | **93.96** | **0.54** | **103.44** | **5.53** | - | **16.62** | **57.18** | **81.03** | **41.88** | **9.86** | **24.08** | **2.30** | - | **5.16** | **46.11** | **77.04** |
| **P value** | **.0001** | **0.71** | **.0001** | **0.01** | - | **0.00** | **.0001** | **.0001** | **.0001** | **0.00** | **.0001** | **0.13** | - | **0.02** | **.0001** | **.0001** |
| **CV (%)** | **5.8** | **18** | **6.4** | **28** | - | **1.3** | **4.1** | **2.7** | **5.8** | **1.3** | **6.5** | **16** | - | **1.7** | **3.4** | **1.8** |

Where:- Means with column followed by the same letter are not significantly different from each other at P ≤ 0.0, LUT = Land Use Types, NFL = Natural Forest Land, GzL = Grazing Land, CuL = Cultivated Land, CoL = Coffee Farm Land, EuPL = Eucalyptus Plantation Land, C = Clay, SC = Sandy Clay, SCL = Sandy Clay Loam, Si/C = Silt-Clay ratio, PD = Particle Density, BD = Bulk Density, TP = Total Porosity, LSD = least Significance Difference, CV = Coefficient of Variation.

sand fraction in soil under EuPL and CuL might be due to the selective removal of clay particles by erosion and leaving the sand particles, which ultimately increases the contents of the sand fraction. This finding is in agreement with research carried out in the Shihatig watershed [36] and Chemoga watershed [37] of northern Ethiopia. In brief, based on the sand, silt and clay fraction, the texture of the studied soil varied from sandy clay loam to clay textural class. In line with this finding [21, 38] reported a significant change in soil texture between various LUTs due to different erosion statuses, cover, and tillage activities. The difference in soil texture between land uses shows that the influence of land-use conversion on soil texture is caused by different land utilization and management systems [39].

The soil particle density, bulk density, and soil total porosity varied significantly (p<0.05) under different LUTs (Table 1). The soil bulk density of the NFL was significantly lower than other LUTs. The soil under EuPL had significantly higher bulk density followed by the soil under CuL and GzL. The soil bulk density of the study area ranged from 0.90 g/cm³ found in the soil under NFL to 1.47 g/cm³ attained by the soil under EuPL. The soil particle density ranged from 2.3 g/cm³ found in the soil under NFL and CoFL to 2.6 g/cm³ obtained by the soil under GzL.

The soil bulk density was higher in EuPL because of the continuous exposure of soil to the direct impact of raindrops leads to soil compaction. Soil compaction resulting from

overgrazing might have caused higher bulk density values in the soil under GzL compared to CuL, NFL and CoFL. Additionally, the high soil bulk density in CuL due to intensive tillage practices that may temporarily compact the tilled soil layer and the relatively lower soil organic matter content due to the removal of crop residues. However, the observed low bulk density in soils under the NFL and CoFL could largely be attributed to their high soil organic matter content. The correlation analysis data also showed a negative relationship ($r = -0.88^{**}$) between soil bulk density and organic matter of studied soil consistence with the work of [14, 35, 40]. Additionally, inadequate soil aggregation may be the cause of the higher soil bulk density beneath the cultivated and eucalyptus trees than in the forestlands [21, 36]. The soil bulk density values found in the present study area fall within the typical ranges obtained in the majority of mineral soils. The soil bulk density fell within the expected range, which means that water movement and aeration within the soil structure are in a favourable state to support plant growth and govern the quantity and diversity of soil microbes, which have a variety of uses in agricultural activities. The critical bulk density of clay soil is approximately $1.4 \text{ cm}^{-3}$ [41]. Thus, the bulk density of the soil surface in the present study site was within an acceptable range for agricultural purposes.

Depending on soil bulk and particle density, the soil total porosity (TP) of the present study area ranged from 42 to 62%. The highest soil TP (62%) was attained by the soil of NFL followed by CoFL due to comparatively lower animal trampling and high SOM content than that of CuL, GzL and EuPL. The decrease in TP of soil under CuL, GzL and EuPL is closely associated with the degree of soil organic matter loss, which is controlled by the status of soil management methods [22, 42]. Generally, [43] state that to support and control the activities of soil biota, the ideal TP of sand particles was approximately 40%, whereas that of soil with a clay content was approximately 50% and above. Using this as a baseline, the study's findings confirm that there are no problems with soil qualities under the investigated land-use types through soil aeration and water infiltration.

### Effects of land use types on soil chemical properties

**Soil pH and exchangeable acidity.** The soil pH was significantly ($p<0.05$) varied among various land use types in the study area (Table 2). The lowest soil pH (4.90) was found in soil under CuL, whereas the highest pH (5.59) was observed in soil under NFL. Soil pH under NFL and CoFL were numerically high, whereas CuL, GzL and EuPL had lower soil pH levels across all study sites. The lower pH in soil under CuL, GzL and EuPL may be ascribed to the leaching of cations due to high precipitation, removal of cations from croplands through residue harvesting for fuel, construction, and fencing, as well as cation uptake by grazing and eucalyptus plantation without replenishment. In agreement with the current findings, several researchers have reported that soil cations are depleted through crop harvesting, leading to lower soil pH in continuously ploughed lands compared to adjacent land use types [39, 42, 44, 45]. Additionally, eucalyptus trees absorb large amounts of basic soil cations, which are not returned to the soil system rapidly, causing a further reduction in soil pH compared to adjacent land uses. The oblong-shaped canopy of eucalyptus also contributes to this problem, as it produces large raindrops that enhance cations leaching and the release of organic acids during organic matter mineralization, further lowering soil pH [14, 46–48]. The soil pH in the research area varied from very strongly acidic (4.5–5.0) to moderately acidic (5.6–6.0), following the soil pH classification by [43]. CuL, GzL and EuPL had soil pH values in the very strongly acidic classes, suggesting that crop production and microbial activity in these areas may be restricted. Thus, it requires amelioration using materials such as lime, farmyard manure, compost and vermicompost.

**Table 2. Effects of land use changes on selected soil chemical properties of different villages of Sayo district.**

| LUT | pH | ExAc | ExAl | AS | OM | TN | C:N | Av.P | pH | ExAc | ExAl | AS | OM | TN | C:N | Av.P |
|---|---|---|---|---|---|---|---|---|---|---|---|---|---|---|---|---|
| | | cmolcKg$^{-1}$ | | | % | | | (mg/kg) | | cmolcKg$^{-1}$ | | | % | | | (mg/kg) |
| | | | | Aba Jarra | | | | | | | | | Alaku Dorgomme | | | |
| NFL | 5.48$^a$ | 0.6$^e$ | 0.2$^e$ | 6$^d$ | 6.4$^a$ | 0.46$^a$ | 8.1$^d$ | 7$^a$ | 5.51$^a$ | 0.45$^e$ | 0.2$^d$ | 4$^d$ | 7.1$^a$ | 0.41$^a$ | 10$^d$ | 13$^a$ |
| GzL | 5.24$^b$ | 1.7$^c$ | 1.3$^c$ | 28$^b$ | 3.5$^d$ | 0.18$^d$ | 11.1$^b$ | 3.3$^b$ | 5.39$^b$ | 1.76$^c$ | 1$^c$ | 27$^c$ | 3.7$^c$ | 0.16$^c$ | 13$^b$ | 4$^c$ |
| CuL | 4.99$^c$ | 2.5$^a$ | 1.9$^a$ | 33$^a$ | 5.3$^c$ | 0.23$^c$ | 13.1$^a$ | 3.8$^b$ | 4.99$^c$ | 2.08$^b$ | 1.7$^b$ | 34$^b$ | 3.9$^c$ | 0.40$^b$ | 14$^a$ | 3.4$^d$ |
| CoFL | 5.33$^b$ | 1.3$^d$ | 0.8$^d$ | 14$^c$ | 5.9$^b$ | 0.41$^b$ | 8.4$^d$ | 5.7$^{ab}$ | 5.54$^c$ | 0.7$^d$ | 0.3$^d$ | 5$^d$ | 5.9$^b$ | 0.16$^c$ | 9$^e$ | 9$^b$ |
| EuPL | 4.91$^c$ | 2.3$^b$ | 1.8$^b$ | 30$^b$ | 3.2$^d$ | 0.19$^d$ | 9.9$^c$ | 3.7$^b$ | 5.09$^c$ | 2.5$^a$ | 2.0$^a$ | 39$^a$ | 3$^d$ | 0.15$^d$ | 12$^c$ | 2.5$^e$ |
| **Mean** | **5.19** | **1.68** | **1.17** | **22** | **4.83** | **0.29** | **10.13** | **4.74** | **5.30** | **1.50** | **1.06** | **21** | **4.71** | **0.26** | **11.6** | **6.31** |
| LSD $_{(0.05)}$ | **0.1** | **0.2** | **0.11** | **3.3** | **0.3** | **0.04** | **1.24** | **2.73** | **0.1** | **0.2** | **0.1** | **3.8** | **0.4** | **0.04** | **0.9** | **0.4** |
| F | 24.97 | 366.48 | 152.78 | 118.57 | 183.44 | 110.18 | 27.67 | 3.43 | 43.76 | 326.30 | 190.17 | 180.77 | 178.75 | 108.42 | 55.64 | 776.51 |
| P value | .0001 | .0001 | .0001 | .0001 | .0001 | .0001 | .0001 | 0.05 | .0001 | .0001 | .0001 | .0001 | .0001 | .0001 | .0001 | .0001 |
| CV (%) | 1.6 | 6.4 | 5.5 | 8.3 | 3.8 | 7.2 | 6.7 | 31 | 1.2 | 7.4 | 7.5 | 9.7 | 4.6 | 8.8 | 4.5 | 4.3 |
| | | | | Abichu Shogo | | | | | | | | | Tabor | | | |
| NFL | 5.59$^a$ | 0.35$^e$ | 0.1$^d$ | 2$^e$ | 7.7$^a$ | 0.6$^a$ | 8$^c$ | 14$^a$ | 5.45$^a$ | 0.57$^c$ | 0.29$^c$ | 6$^c$ | 5.7$^a$ | 0.41$^a$ | 8$^c$ | 10$^a$ |
| GzL | 5.03$^c$ | 2.94$^a$ | 1.77$^a$ | 43$^a$ | 3.2$^d$ | 0.1$^d$ | 13$^a$ | 3.3$^d$ | 5.24$^b$ | 1.76$^b$ | 1.24$^b$ | 20$^b$ | 4.7$^c$ | 0.28$^b$ | 9$^{bc}$ | 4.3$^c$ |
| CuL | 5.26$^b$ | 1.97$^c$ | 1.48$^b$ | 22$^c$ | 4$^c$ | 0.2$^c$ | 11$^b$ | 4.3$^c$ | 4.9$^d$ | 2.17$^a$ | 1.84$^a$ | 27$^a$ | 4.4$^c$ | 0.24$^{bc}$ | 11$^{ab}$ | 4.4$^c$ |
| CoFL | 5.53$^a$ | 0.8$^d$ | 0.36$^c$ | 9$^d$ | 5.8$^b$ | 0.4$^b$ | 9$^c$ | 6.7$^b$ | 5.39$^a$ | 0.57$^c$ | 0.29$^c$ | 6$^c$ | 5.3$^b$ | 0.28$^b$ | 11$^{ab}$ | 5.7$^b$ |
| EuPL | 5.24$^b$ | 2.26$^b$ | 1.59$^{ab}$ | 30$^b$ | 3.2$^d$ | 0.2$^{cd}$ | 12$^b$ | 3.4$^d$ | 5.13$^c$ | 1.88$^b$ | 1.7$^a$ | 25$^a$ | 3.9$^d$ | 0.18$^c$ | 13$^a$ | 3.1$^d$ |
| **Mean** | **5.33** | **1.66** | **1.05** | **21** | **4.78** | **0.29** | **10.6** | **6.51** | **5.22** | **1.39** | **1.09** | **17** | **4.8** | **0.28** | **10.5** | **5.42** |
| LSD $_{(0.05)}$ | **0.14** | **0.1** | **0.2** | **2.9** | **0.4** | **0.05** | **1.1** | **0.3** | **0.09** | **0.2** | **0.2** | **3.4** | **0.4** | **0.06** | **2.1** | **0.7** |
| F | 24.57 | 120.89 | 514.27 | 306.64 | 225.72 | 113.92 | 32.12 | 1629.66 | 51.31 | 123.70 | 105.53 | 86.01 | 35.20 | 18.08 | 5.98 | 117.06 |
| P value | .0001 | .0001 | .0001 | .0001 | .0001 | .0001 | .0001 | .0001 | .0001 | .0001 | .0001 | .0001 | .0001 | 0.00 | 0.01 | .0001 |
| CV (%) | 1.5 | 4.8 | 11.4 | 7.5 | 4.6 | 9.8 | 6 | 3.1 | 1 | 9.2 | 10 | 11 | 4.4 | 12.4 | 11 | 7.5 |

Where:- Means with column followed by the same letter are not significantly different from each other at P ≤ 0.05, LUT = Land Use Types, NFL = Natural Forest Land, GzL = Grazing Land, CuL = Cultivated Land, CoL = Coffee Farm Land, EuPL = Eucalyptus Plantation Land, AS = Acid Saturation, Ex Aci = Exchangeable acidity, Ex Al = Exchangeable Aluminium, OM = Organic Matter, TN = Total Nitrogen, Av. P = Available Phosphorous, CV = Coefficient of Variation, LSD = Least Significance Difference.

The soil exchangeable aluminium, exchangeable acidity, and acid saturation percentage of the study area were significantly (p<0.05) affected by land use types (Table 2). The soil under NFL had the lowest exchangeable acidity compared to adjacent land use types. The soil under GzL had the highest (2.94 cmolc kg$^{-1}$) soil exchangeable acidity followed by EuPL and CuL (2.5 cmolc kg$^{-1}$) lands. The soils under the NFL and CoFL had lower exchangeable acidity, however, those of CuL, GzL and EuPL showed numerically higher exchangeable acidity due to less soil pH. The lower exchangeable acidity in forest soils may result from organic matter accumulation due to minimal soil disturbance. Exchangeable aluminum followed a similar pattern with exchangeable acidity, hence the lowest mean value (0.1 cmolc kg$^{-1}$) was found in soil under NFL and the highest (2 cmolc kg$^{-1}$) was found in soil under EuPL. The lower soil pH levels contributed to higher exchangeable aluminium, whereas higher pH corresponded to lower aluminium levels. This is consistent with a report by [49], who noted that a pH drop below 5.5 increases the solubility of Al-containing minerals, raising the possibility of aluminium toxicity in plants. In brief, the higher exchangeable acidity and aluminium observed in eucalyptus, grazing, and cultivated lands is likely due to continuous cultivation and the loss of essential cations through plant uptake. This suggests a higher concentration of free $Al^{3+}$ in the soil solutions of these converted lands compared to forest and coffee farm soils agreed with studies by [21, 50].

Land use conversion also aggravates the percentages of acid saturation. The significantly highest (43%) and lowest (2.0%) average soil acid saturation percentages were found in the soil under GzL and NFL, respectively. Relatively, the soil of GzL, CuL and EuPL obtained numerically higher AS percentages than the soil of NFL and CoFL. The present finding is consistence with [24], and [54] who reported high AS percentages under EuPL and CuL because of low soil pH and high exchangeable acidity as the result of frequent use of inorganic fertilizers and leaching of cations. According to [43], the average exchangeable acidity in study sites is categorized as very high under the soil EuPL, CuL and GzL.

**Soil organic matter, total nitrogen and carbon to nitrogen ratio.** The results showed that land use types significantly (p<0.05) affected soil organic matter (SOM) content at the research sites (Table 2). The highest SOM (7.7%) was found in soils under NFL, whereas the lowest SOM (3%) was obtained in soils from EuPL. This indicates that converting NFL has led to a significant reduction in SOM. The least significant difference (LSD) test further confirmed that SOM levels in EuPL, GzL, and CuL were significantly lower (p < 0.05) than in soils NFL and CoFL. The relatively higher soil organic matter in NFL and CoFL can be attributed to the accumulation of plant residues and less disturbance compared to other land use types. Additionally, sufficient soil moisture due to adequate rainfall and vegetation cover in these areas promotes the growth of various herbs and grasses, leading to slower decomposition and higher SOM levels. Consistent with these findings, previous studies have reported significantly lower SOM in cultivated land compared to adjacent land uses due to continuous cultivation and the complete removal of crop residues for livestock feed, fuel, construction, and fencing purposes [51–53].

Comparing soil organic in crop and coffee farmland, [54] found significantly higher organic matter in coffee farmlands, likely attributable to dense canopy, minimal soil disturbance, and slower organic matter decomposition caused by cooler soil temperatures from shading. In contrast, in croplands, higher decomposition rates and the total removal of crop biomass from the field caused the lower SOM content. Furthermore, several studies [14, 46, 55] reported significantly lower SOM in eucalyptus plantations compared to adjacent forest, grazing land, and cultivated lands due to the slow decomposition rate of eucalyptus leaves and the collection of debris for fuel, which reduces organic matter accumulation under the trees. According to [43] classification for assessing soil health, SOM levels in GzL, EuPL and CuL are considered moderate, while soils under natural forests and coffee farmlands are categorized as having very high SOM levels.

The indicated a significant (p<0.05) variation in soil total nitrogen (TN) between various land use types (Table 2). The lowest TN (0.1%) was recorded in GzL soils, whereas the highest TN (0.6%) was found in NFL soils. Soils under NFL contained the highest TN levels compared to other land uses, likely because of the continuous accumulation of OM and minimal soil disturbance. The high microbial activity in NFL soils also promotes nitrogen mineralization and nutrient cycling. The soil under NFL and CoFL showed a higher TN than other land use types. This may be ascribed to the year-round vegetation cover, which enhances SOM. In NFL, nitrogen-fixing trees, decaying plant matter, and leaf litter contribute to higher organic matter, thereby increasing soil TN. The correlation analysis showed that the TN had a strong positive association with SOM (0.94**) (Table 4). The findings suggest that TN content declines when natural forests are converted into cultivated, grazing, or plantation lands, particularly eucalyptus plantation land. Several researchers reported reductions in TN following land use changes and have emphasized its strong positive correlation with soil organic matter [46, 52, 56]. According to [43], the TN levels in the research area was categorized as low (0.05–0.15%) to moderate (0.15–0.25%) in the soil under CuL, GzL, and EuPL, whereas NFL and CoFL soils were classified as high (0.25–0.50% and above). Therefore, to boost TN and improve land

productivity, crop yields, and biodiversity, especially in cultivated, grazing, and eucalyptus lands; additional sources of nitrogen, particularly eco-friendly organic fertilizers like farmyard manure, compost, vermicompost, and bio-fertilizers, are recommended.

The C/N ratio reflects the balance between carbon and nitrogen in organic materials and is affected by soil organic carbon and TN content. The results indicated that the C/N ratio significantly varied among land use types in the study area. The highest C/N ratio (14:1) was found in soil under CuL, whereas the lowest C/N ratio (8:1) was obtained by soil under NFL (Table 2). The higher C/N ratios in CuL and GzL compared to other land uses can be ascribed to the relatively low soil TN content compared to soil OC in these areas. A narrow C/N ratio indicates efficient mineralization, whereas a wide ratio suggests nitrogen immobilization. The greater mineralization observed in soil under NFL and CoFL may because of the higher root biomass and turnover in these permanent vegetation areas, which enhances SOM contents. Several researches reported significantly lower C/N ratios in natural forests [57] and coffee farmland soil [54], which is agreed with the findings of this study. Based on the rating system by [43], the soil C/N ratios in the current study area ranged from very low (<10:1) to low (10–15:1) for organic materials in the surface soil. When the C/N ratio exceeds 30:1, nitrogen becomes immobilized by soil microorganisms. In contrast, when the ratio is below 20:1, mineral nitrogen is released into the soil system. Since the C/N ratios in the research site was below 20:1, it indicates that mineral nutrients are being released into the soil and plant environment, suggesting that no critical problems exist regarding the C/N ratio [41].

**Available phosphorous.**   The soil available phosphorus (Av. P) was significantly ($p < 0.05$) influenced by land use conversion, showing remarkable differences across various land use types (Table 2, (S2 Table). The soil under NFL had the highest Av. P, followed by CoFL, whereas EuPL soils showed the significantly lowest Av. P values. The comparatively high Av. P in the soil of NFL can be ascribed to the high organic matter content, which releases organic phosphorus resulting in increased availability of phosphorus in the soil. In the acidic soils of the study site, higher pH likely contributed to the precipitation of aluminium, iron, and manganese in soil solution, reducing phosphorus fixation and increasing its availability. This reflects a positive correlation of available phosphorus with organic matter (r = 0.86**), and soil pH (r = 0.71*), which agreed with the works of [21, 36]. On the other hand, the low available phosphorus in eucalyptus plantation land soils may be due to the trees' ability to immobilize phosphorus, limiting its availability for plant uptake, as stated by [58]. According to the classification system by [43], soil Av. P in the study site varied from very low (<5 mg/kg) in GzL, EuPL, and CuL, low (5–10 mg/kg) in CoFL, and low (5–10 mg/kg) to moderate (10–15 mg/kg) in soil under NFL. The acidic nature of soil may increase the solubility of heavy metals like aluminium and iron, which can fix phosphorus and reduce its availability. Thus, the soil fertility of the study area, particularly in GzL, EuPL, and CuL, may decline due to low available phosphorus levels. Therefore, it is necessary to apply additional phosphorus sources, including farmyard manure, poultry manure, compost, vermicompost, inorganic phosphorus fertilizers, and liming materials, to reduce phosphorus fixation in acidic soils and enhance fertility.

**Exchangeable cations ($Ca^{2+}$, $Mg^{2+}$ & $K^+$)..**   The results showed that the exchangeable cations ($Ca^{2+}$, $Mg^{2+}$, and $K^+$) were significantly ($p<0.05$) affected by land use types (Table 3). The soil under NFL had the highest exchangeable calcium ($Ca^{2+}$) levels, whereas GzL soils showed the lowest exchangeable calcium. Similarly, the highest exchangeable magnesium ($Mg^{2+}$) and potassium ($K^+$) were found in NFL soils, while the lowest values were observed in soil under EuPL. The mean values of exchangeable calcium, magnesium, and potassium ranged from 2.2–8, 1.1–3.7, and 0.3–1.2 cmolc $kg^{-1}$, respectively (S2 Table). Soils from NFL and CoFL had higher levels of exchangeable cations compared to soil under GzL, EuPL, and CuL. This difference is likely because of the addition of organic matter through the build-up of plant residues,

**Table 3. Effects of land use changes on selected soil physical properties of Sayo district.**

| LUT | Ca | Mg | K | CEC | PBS | Fe | Mn | Zn | Cu | Ca | Mg | K | CEC | PBS | Fe | Mn | Zn | Cu |
|---|---|---|---|---|---|---|---|---|---|---|---|---|---|---|---|---|---|---|
| | cmolc kg⁻¹ | | | | (%) | (mg/kg) | | | | cmolc kg⁻¹ | | | | (%) | (mg/kg) | | | |
| | Aba Jarra | | | | | | | | | Alaku Dorgomme | | | | | | | | |
| NFL | 6[a] | 2.6[a] | 0.82[a] | 34[a] | 30[b] | 25[d] | 30[b] | 2.2[d] | 1.3[c] | 7.4[b] | 3.2[b] | 1[a] | 33[a] | 36[b] | 23[b] | 26[c] | 2[c] | 1.16[c] |
| GzL | 2.4[e] | 1.3[d] | 0.62[b] | 20[c] | 23[c] | 42[b] | 29[b] | 3.1[b] | 2[b] | 2.9[c] | 1.4[c] | 0.4[c] | 20[b] | 24[c] | 46[a] | 41[b] | 4[b] | 2.14[b] |
| CuL | 3.4[c] | 1.15[e] | 0.49[b] | 17[d] | 30[ab] | 45[ab] | 38[a] | 3.5[a] | 2.5[a] | 2.3[d] | 1.2[c] | 0.5[c] | 23[b] | 18[d] | 51[a] | 39[b] | 6[a] | 2.07[b] |
| CoFL | 5[b] | 2.1[b] | 0.64[b] | 25[b] | 32[a] | 33[c] | 36[ab] | 3.2[b] | 1.5[c] | 8[a] | 3.5[a] | 1.1[a] | 33[a] | 40[a] | 25[b] | 29[c] | 3[b] | 1.24[c] |
| EuPL | 3.2[d] | 1.7[c] | 0.54[b] | 19[cd] | 27[b] | 47[a] | 36[ab] | 2.7[c] | 1.3[c] | 2.4[d] | 1.1[c] | 0.3[d] | 20[b] | 20[d] | 46[a] | 48[a] | 5[a] | 2.38[a] |
| **Mean** | **4.07** | **1.76** | **0.62** | **23.06** | **28.4** | **38.4** | **34.2** | **2.94** | **1.73** | **4.57** | **2.08** | **0.66** | **25.5** | **27** | **38.3** | **36.5** | **3.75** | **1.8** |
| **LSD** (0.05) | **0.24** | **0.14** | **0.15** | **2.3** | **2.7** | **5** | **8.2** | **0.2** | **0.3** | **0.22** | **0.3** | **0.12** | **3.4** | **2.2** | **6.2** | **4.3** | **0.3** | **0.22** |
| **F value** | **1579.81** | **137.32** | **87.77** | **38.86** | **188.75** | **44.71** | **43.38** | **181.15** | **59.87** | **1579.81** | **137.32** | **87.77** | **38.86** | **188.75** | **44.71** | **43.38** | **181.15** | **59.87** |
| **P value** | **.0001** | **.0001** | **.0001** | **.0001** | **.0001** | **.0001** | **.0001** | **.0001** | **.0001** | **.0001** | **.0001** | **.0001** | **.0001** | **.0001** | **.0001** | **.0001** | **.0001** | **.0001** |
| **CV (%)** | **3.2** | **4.4** | **13** | **6.2** | **5.2** | **7.2** | **13.2** | **4.5** | **9** | **2.7** | **8.4** | **10** | **7.2** | **4.4** | **8.9** | **6.5** | **3.8** | **6.9** |
| | Abichu Shogo | | | | | | | | | Tabor | | | | | | | | |
| NFL | 8.7[a] | 3.7[a] | 1.2[a] | 38[a] | 39[a] | 22[d] | 25[c] | 1.5[d] | 1[c] | 6[a] | 2.6[a] | 0.8[a] | 36[a] | 27[b] | 22[b] | 34[ab] | 2.3[c] | 1.3[b] |
| GzL | 2.2[e] | 1.2[d] | 0.4[c] | 20[c] | 19[c] | 46[a] | 42[a] | 3.6[b] | 2.3[a] | 4[c] | 2.2[ab] | 0.6[ab] | 26[b] | 27[b] | 38[ab] | 44[a] | 3.4[b] | 1.2[b] |
| CuL | 3.6[c] | 2.2[b] | 0.8[b] | 27[b] | 25[b] | 42[ab] | 28[bc] | 4.1[a] | 1.9[b] | 3.2[d] | 2.1[b] | 0.5[b] | 27[b] | 22[c] | 41[a] | 36[ab] | 4[a] | 2.1[a] |
| CoFL | 5.2[b] | 2.2[b] | 0.8[b] | 36[a] | 24[b] | 33[c] | 28[bc] | 2.1[c] | 1.1[c] | 5.3[b] | 2.2[ab] | 0.7[ab] | 28[b] | 30[a] | 29[ab] | 33[b] | 3[b] | 1.2[b] |
| EuPL | 3.2[d] | 1.6c | 0.5[c] | 21[c] | 25[b] | 37[bc] | 32[b] | 3.7[ab] | 1.3[c] | 3.5[d] | 1.4c | 0.5[b] | 22c | 25[b] | 40[a] | 35[ab] | 3[b] | 1.3[b] |
| **Mean** | **4.55** | **2.37** | **0.78** | **28.4** | **26.5** | **36** | **30.08** | **3.04** | **1.53** | **4.40** | **2.11** | **0.66** | **27.7** | **26.2** | **33.7** | **36.4** | **3.18** | **1.41** |
| **LSD** (0.05) | **0.4** | **0.2** | **0.1** | **3.4** | **1.8** | **5.4** | **4.7** | **0.4** | **0.3** | **0.2** | **0.3** | **0.2** | **3.5** | **2.1** | **17.5** | **10.6** | **0.4** | **0.2** |
| **F value** | **301.98** | **305.26** | **29.14** | **55.85** | **167.10** | **27.38** | **19.71** | **60.08** | **29.35** | **165.27** | **13.38** | **3.49** | **19.12** | **18.70** | **2.29** | **1.57** | **16.98** | **29.29** |
| **P value** | **.0001** | **.0001** | **.0001** | **.0001** | **.0001** | **.0001** | **.0001** | **.0001** | **.0001** | **.0001** | **0.00** | **0.05** | **0.00** | **0.00** | **0.13** | **0.26** | **0.00** | **.0001** |
| **CV (%)** | **5.5** | **5.7** | **13** | **6.5** | **3.7** | **8.3** | **8.4** | **8.4** | **11.4** | **3.6** | **9.3** | **16.5** | **6.9** | **4.5** | **28.5** | **16** | **7.8** | **8.3** |

Where:- Means with column followed by the same letter are not significantly different from each other at P ≤ 0.05, LUT = Land Use Types, NFL = Natural Forest Land, GzL = Grazing Land, CuL = Cultivated Land, CoL = Coffee Farm Land, EuPL = Eucalyptus Plantation Land, CV = Coefficient of Variation, LSD = Least Significance Difference, CEC = Cation Exchange Capacity, PBS = Percentage of Base Saturation, SEB = Sum of Exchangeable Bases.

leaves, and litter in NFL and CoFL, which is less prevalent in CuL, GzL, and EuPL. The sum exchangeable cations of the study area had a significant positive correlation to soil organic matter content (r = 0.90**) (Table 4). This finding aligns with the works of [34, 35, 42, 59] that reported significantly higher exchangeable cations in soil under NFL due to organic matter accumulation.

The higher exchangeable cation levels in coffee farmlands compared to crops, grazing, and eucalyptus lands may result from organic matter deposition, minimal soil disturbance, reduced leaching, less erosion, and better soil management practices in coffee farmlands, as reported by [54]. Conversely, lower exchangeable cations in eucalyptus land soils, as reported by [46], could be due to lower soil pH, which increases the solubility of Fe, Mn and Al, replacing basic cations, along with the absorption of large amounts of cations by eucalyptus that are not easily returned to the soil. Based on [60] ratings, the content of soil exchangeable calcium in the study area was categorized as low (2–5 cmolc kg⁻¹) in EuPL, GzL, and CuL, and medium (5–10 cmolc kg⁻¹) in NFL and CoFL. Exchangeable magnesium was classified as medium (1–3 cmolc kg⁻¹) in most LUTs, except in NFL and CoFL of Alaku village, which had high concentrations (3–8 cmolc kg⁻¹) whereas the soil exchangeable potassium concentrations of the study area ranged from medium (0.3–0.6 cmolc kg⁻¹) to high (0.6–1.2 cmolc kg⁻¹). In converted

**Table 4. The correlation coefficient between soil physiochemical properties of the studied area.**

|  | Sand | Clay | BD | pH | OM | TN | Av. P | ExAl | ExAc | AS | SEB | CEC | PBS | Fe | Mn | Zn |
|---|---|---|---|---|---|---|---|---|---|---|---|---|---|---|---|---|
| **Sand** | 1 | | | | | | | | | | | | | | | |
| **Clay** | -.93** | 1 | | | | | | | | | | | | | | |
| **BD** | .82** | -.90** | 1 | | | | | | | | | | | | | |
| **pH** | -.68* | .73* | -.69* | 1 | | | | | | | | | | | | |
| **OM** | -.85** | .92** | -.88** | .72* | 1 | | | | | | | | | | | |
| **TN** | -.87** | .93** | -.89** | .75** | .94** | 1 | | | | | | | | | | |
| **Av. P** | -.85** | .86** | -.76** | .71* | .86** | .85** | 1 | | | | | | | | | |
| **ExAl** | .83** | -.87** | .85** | -.91** | -.83** | -.86** | -.80** | 1 | | | | | | | | |
| **ExAc** | .84** | -.89** | .86** | -.86** | -.84** | -.85** | -.82** | .96** | 1 | | | | | | | |
| **AS** | .82** | -.90** | .88** | -.84** | -.87** | -.88** | -.85** | .94** | .97** | 1 | | | | | | |
| **SEB** | -.84** | .88** | -.84** | .78** | .90** | .91** | .91** | -.86** | -.86** | -.90** | 1 | | | | | |
| **CEC** | -.80** | .86** | -.86** | .77** | .82** | .87** | .80** | -.85** | -.86** | -.87** | .87** | 1 | | | | |
| **PBS** | -.64* | .68* | -.62* | .58* | .75** | .72* | .74* | -.65* | -.65* | -.73* | .84** | .50* | 1 | | | |
| **Fe** | .72* | -.79** | .81** | -.78** | -.78** | -.81** | -.76** | .83** | .83** | .86** | -.84** | -.81** | -.66* | 1 | | |
| **Mn** | .51* | -.54* | .51* | -.61* | -.54* | -.54* | -.57* | .60* | .63* | .65* | -.63* | -.60* | -.50* | .65* | 1 | |
| **Zn** | .63* | -.71* | .63* | -.65* | -.71* | -.75** | -.71* | .74* | .71* | .73* | -.71* | -.66* | -.60* | .71* | .61* | 1 |
| **Cu** | .52* | -.61* | .65* | -.64* | -.57* | -.64* | -.57* | .69* | .73* | .78** | -.71* | -.68* | -.58* | .72* | .60* | .71* |

NB:

* Correlation is significant at the 0.05 level (2-tailed).

** Correlation is significant at the 0.01 level (2-tailed).

BD = bulk density, OM = organic matter, TN = total nitrogen, Av. P = available phosphorous, ExAl = exchangeable aluminium, ExAc = exchangeable acidity, AS = acid saturation, CEC = cation exchangeable

lands with low exchangeable cations, soil amendments are required to enhance soil fertility and crop productivity. Thus, liming materials, for example lime, gypsum, biochar, dolomite, ash, and other organic fertilizers are recommended to enhance exchangeable cations, as suggested by [35, 61–64].

**Soil cations exchangeable capacity and percentage of base saturation.** The output of analysed soil indicated that, the cation exchange capacity (CEC) was significantly ($p < 0.05$) varied across the adjacent land use types in the investigated villages (Table 3). Accordingly, highest CEC (38 cmolc kg$^{-1}$) was found under the soil of NFL, whereas the lowest (17 cmolc kg$^{-1}$) was recorded in soil under CuL. In comparison, the soil under NFL and CoFL showed numerically higher CEC, whereas the soil under GzL, CuL, and EuPL had significantly lower CEC due to their less organic matter content and clay fraction. The correlation analysis of soil physicochemical properties of the present study showed that the CEC had significant positive relationship with soil clay particle (r = 0.86**) and SOM (r = 0.82**) (Table 4). This agreed with findings of (22, 47, 75), which reported a positive correlation between organic matter content and CEC. Additionally, (76) emphasized that factors like deforestation, leaching, less recycling of organic materials, shorter fallow periods, and soil erosion contribute to the reduction of basic cations subsequently for soil CEC in CuL compared to the soil under NFL. According to (70), the soil in the research site was generally rated as medium (12–25 cmol (+) kg$^{-1}$) to high (25–40 cmol (+) kg$^{-1}$) CEC levels.

The percentage of base saturation (PBS) was significantly ($p < 0.05$) influenced by conversion in land uses (Table 3). Hence, the highest PBS was found in CoFL soils, whereas CuL had the lowest PBS. The soil under NFL and CoFL had numerically higher PBS compared to EuPL,

GzL, and CuL due to higher organic matter, high exchangeable cations with the higher soil pH of respective site. This finding is agreement with the work of [62], who reported that soils with higher levels of exchangeable calcium and magnesium typically show higher PBS. According to [60], the PBS levels in the study area were categorized as very low (0–20%) to low (20–40%). The PBS levels also reflect the degree of leaching in the soils, revealing that the studied villages' soils were exposed to significant leaching, with PBS ranging from severely leached (15–30%) to moderately leached (30–50%). This implies that the soils are heavily degraded and require immediate intervention. Several researchers recommend management strategies such as lime application, adding organic matter, mulching, nutrient management, crop rotation, and soil erosion control measures to address these issues [35, 61–64]. These practices are essential for improving soil fertility and maintaining the productivity of the land.

**Soil extractable micronutrients (Fe, Mn, Zn and Cu).** The extractable Fe, Mn, Zn, and Cu were significantly ($p < 0.05$) influenced by changes in land use (Table 3, S2 Table). The highest concentration of Fe (51 mg/kg) was found in CuL soils whereas the lowest Fe (22 mg/kg) was obtained in soil under NFL. The highest Mn (48 mg/kg) was observed in soil under EuPL whereas the lowest (25 mg/kg) was found in the soil under NFL. The highest Zn (6 mg/kg) levels were registered in the soil under CuL and the lowest (1.5 mg/kg), were found in NFL soil whereas the highest Cu (2.5 mg/kg) levels were found in soil under CuL and lowest Cu (1.0 mg/kg) in soil under NFL. Comparing the various land uses, soils under NFL and CoFL showed numerically lower extractable micronutrient levels than those under CuL, EuPL and GzL. This difference can be ascribed to the higher SOM, higher soil pH, and lower soil exchangeable acidity in NFL and CoFL. These findings align with the works of [39, 59, 65], which reported lower micronutrient levels in forest soils compared to adjacent land use types, likely because of the higher soil pH that decreases the solubility and availability of these elements. Based on the ratings established by [66], extractable Fe, Mn, and Zn levels were categorized as high, however, Cu was rated as low (0.3–2.5 mg/kg) in the soil of the study area. The higher average values of these micronutrients can hinder the availability of essential macronutrients, such as phosphorus and nitrogen, in addition to exchangeable cations, particularly in acidic soils. Therefore, strategic management methods are necessary to alleviate the toxicity of these elements. Several researchers recommended practices including the use of lime, biochar, gypsum, and organic fertilizers, which can effectively amend acidic soils and decrease the toxicity of Fe, Zn, and Mn [62, 67].

## Conclusion

The study revealed significant variation in soil physical and chemical properties among land use types. The soil under NFL and CoFL had higher clay fraction, total porosity, pH, organic matter, available phosphorus, exchangeable cations, cation exchange capacity and percentage of base saturation. In contrast, CuL, GzL and EuPL showed higher sand content, bulk density, acidity, and exchangeable aluminium but lower pH and some nutrient levels. The soil fertility depletion and acidity in CuL, GzL and EuPL are linked to continuous cultivation, high rainfall, overgrazing, and acidifying fertilizers, resulting in reduced soil productivity and ecosystem degradation. Sustainable practices like the application of lime, gypsum, biochar, and combined inorganic and organic fertilizers are required to enhance soil health and agricultural sustainability.

## Supporting information

**S1 Table. Climatic data of the study area.**
(XLS)

**S2 Table. Summary statistics of analyzed soil data.**
(XLS)

## Acknowledgments

The authors are thankful to Dambi Dollo University College of Agriculture and Natural Resource and Hawassa University School of Plant and Horticultural Science for their support. Next, they thank all the professionals who assisted with soil sampling and analyses in the laboratory.

## Author Contributions

**Conceptualization:** Abu Regasa, Wassie Haile, Girma Abera.

**Data curation:** Abu Regasa, Wassie Haile, Girma Abera.

**Formal analysis:** Abu Regasa.

**Funding acquisition:** Abu Regasa.

**Investigation:** Abu Regasa, Wassie Haile, Girma Abera.

**Methodology:** Abu Regasa, Wassie Haile, Girma Abera.

**Project administration:** Abu Regasa, Wassie Haile, Girma Abera.

**Resources:** Abu Regasa, Wassie Haile, Girma Abera.

**Software:** Abu Regasa.

**Supervision:** Wassie Haile, Girma Abera.

**Validation:** Abu Regasa, Girma Abera.

**Visualization:** Abu Regasa, Girma Abera.

**Writing – original draft:** Abu Regasa.

**Writing – review & editing:** Abu Regasa, Wassie Haile, Girma Abera.

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
