## [Decision Letter · Decision Letter 0]

10 Sep 2024

PONE-D-24-23222Soil Acidity and Fertility Status of Surface Soils under Different Land Uses in Sayo District of Oromia, Western EthiopiaPLOS ONE

Dear Dr. Regasa,

Thank you for submitting your manuscript to PLOS ONE. After careful consideration, we feel that it has merit but does not fully meet PLOS ONE’s publication criteria as it currently stands. Therefore, we invite you to submit a revised version of the manuscript that addresses the points raised during the review process.

We look forward to receiving your revised manuscript.

Kind regards,

Muhammad Nauman Ahmad, PhD

Academic Editor

PLOS ONE

“The authors are thankful to Dambi Dollo University College of Agriculture and Natural Resource and Hawassa University School of Plant and Horticultural Science for their support in financing. Next, they thank to all the professionals who assisted with soil sampling and analyses in the laboratory.”

Reviewers' comments:

Reviewer's Responses to Questions

**Comments to the Author**

1. Is the manuscript technically sound, and do the data support the conclusions?

Reviewer #1: Partly

2. Has the statistical analysis been performed appropriately and rigorously? 

Reviewer #1: No

3. Have the authors made all data underlying the findings in their manuscript fully available?

Reviewer #1: Yes

4. Is the manuscript presented in an intelligible fashion and written in standard English?

Reviewer #1: No

5. Review Comments to the Author

Reviewer #1: • In the abstract section, there is editorial error and repletion of words that make not encourage reader to read the full article.

• Introduction part is shallow and unable to describe the research gaps and do not shows the problem of conversion of forest land to other LUT.

• The manuscript contains a lot of editorial errors such as poor grammar, punctuation and coherence Please correct it accordingly.

• The ANOVA table should be presented with the appropriate source of variation.

• The results, discussion and conclusion sections are poor, not properly interpreted and justified.

• The critical value of all soil nutrients should be discussed based on the critical value each individual.

• Please correct the subtitle only uppercase the first keyword.

6. PLOS authors have the option to publish the peer review history of their article (what does this mean?). If published, this will include your full peer review and any attached files.

Reviewer #1: **Yes: **Ewunetie Melak

---

## [Author Response · Author response to Decision Letter 0]

6 Nov 2024

College of Agriculture and Natural Resource

Dambi Dollo University

Dambi Dollo City, Ethiopia

Dear reviewers and Editor,

We are thankful for your kind comments on our manuscript entitled “Soil acidity and fertility status of surface soils under different land uses in Sayo district of Oromia, western Ethiopia” following our submission for publication in PLOSE ONE journal. We have re-looked at the manuscript and changed it in line with your comments as follows.

We have incorporated your comments on the abstract and revised it to enhance its attractiveness. Based on your suggestions, we revised the introduction, including the statement of the problem by clearly setting the research gap. Also, we added several new sentences to improve the strength of the introduction. There were no changes made to the materials and methods section, as we did not receive any recommendations regarding the use of specific statistical methods. Therefore, we decided to continue with the methods previously employed unless further suggestions are provided. In response to your comments, we revised the results and discussion sections, ensuring the interpretation aligns with our findings and is supported by recent literature. Additionally, we have corrected grammatical and technical errors to ensure consistency throughout the discussion. Finally, we have modified the conclusion part of the manuscript by providing your constructive comments.

For the editor, the majority of your comments were based on the author's instructions, thus we took all necessary steps to improve the manuscript. Overall, what we want to inform you by using this opportunity is that we updated, rephrased and changed more than half of the manuscript. We respectfully request that you consider this modified text for publication. 

Sincerely,

Abu Regasa (PhD candidate)

Dambi Dollo University

---

## [Decision Letter · Decision Letter 1]

25 Nov 2024

PONE-D-24-23222R1Soil Acidity and Fertility Status of Surface Soils under Different Land Uses in Sayo District of Oromia, Western EthiopiaPLOS ONE

Dear Dr. Regasa,

Thank you for submitting your manuscript to PLOS ONE. After careful consideration, we feel that it has merit but does not fully meet PLOS ONE’s publication criteria as it currently stands. Therefore, we invite you to submit a revised version of the manuscript that addresses the points raised during the review process.

We look forward to receiving your revised manuscript.

Kind regards,

Dafeng Hui, Ph.D.

Academic Editor

PLOS ONE

Journal Requirements:

Additional Editor Comments:

Associate Editor:

The manuscript has been evaluated by the previous reviewer who still has some technical concerns. For example, the ANOVA results were not presented in the main text. I noticed the ANOVA results in supplemental Excel files, but the results need to be summarized and included in one table, or add the information (F and significance) to Tables 1 to 3. A few other editorial issues were found that need to be addresses:

1) grazing should be grazing grassland in several places. Grazing is not a land use type. Add plantation or land after eucalyptus in the abstract (Page 1, line 6).

2) Page 2, 2 paragraph: Delete "But," and start with "Despite these ..."

3) Page 3, the last paragraph: Too much arguments were given but not enough information for the present study. Please add specific objectives or scientific hypotheses here.

4) Page 5, Site selection and soil sampling: how ere natural forest land defined?

5) Page 7, Statistical analysis: physiochemical here and many other places. physiochemical should be changed physicochemical.

7) Page 7, Results and Discussions: change discussions to discussion.

8) Page 8, 2nd paragraph: change to "The results of ANOVA showed that ...". Please also define ANOVA when first time mention in the Data analysis.

Reviewers' comments:

Reviewer's Responses to Questions

**Comments to the Author**

1. If the authors have adequately addressed your comments raised in a previous round of review and you feel that this manuscript is now acceptable for publication, you may indicate that here to bypass the “Comments to the Author” section, enter your conflict of interest statement in the “Confidential to Editor” section, and submit your "Accept" recommendation.

Reviewer #1: (No Response)

2. Is the manuscript technically sound, and do the data support the conclusions?

Reviewer #1: Yes

3. Has the statistical analysis been performed appropriately and rigorously? 

Reviewer #1: Yes

4. Have the authors made all data underlying the findings in their manuscript fully available?

Reviewer #1: Yes

5. Is the manuscript presented in an intelligible fashion and written in standard English?

Reviewer #1: Yes

6. Review Comments to the Author

Reviewer #1: Dear authors: I am happy for most issues are corrected accordingly, but still you mentioned based on ANOVA results. As you state from your manuscript there is no ANOVA table. you have two option, one is to delete the mention of ANOVA from your manuscript , two is to add ANOVA table from your manuscript.

7. PLOS authors have the option to publish the peer review history of their article (what does this mean?). If published, this will include your full peer review and any attached files.

Reviewer #1: **Yes: **Ewunetie Melak

---

## [Author Response · Author response to Decision Letter 1]

3 Dec 2024

Dear Reviewers and Editor,

We would like to express our gratitude for your constructive comments and suggestions on our manuscript entitled “Soil Acidity and Fertility Status of Surface Soils under Different Land Uses in Sayo District of Oromia, Western Ethiopia” submitted for publication in PLOS ONE. Following your feedback, we have thoroughly revised the manuscript to address the points raised, as detailed below:

Journal Requirements and References:

We have addressed the comment regarding retracted articles by removing them from our references. In addition, we have replaced the preprint reference: Regasa A. Conversion in Land Use Change Soil Physiochemical Properties in the Highland of Western Ethiopia, 29 September 2023, PREPRINT (Version 1) available at Research Square [https://doi.org/10.21203/rs.3.rs-3371625/v1]. 2023 with alternative and more suitable citations. We have included one unpublished dataset titled “Kellem Wolega Zone Agriculture and Natural Resource Office. Description of Kellem Wolega Zone Dambi Dollo City, Oromia; 2022” to provide local and contextual data relevant to our analysis.

Supplementary Materials: In line with your suggestion, we have combined the supplementary materials into a single table (S2 Table). This ensures clarity and better alignment with journal requirements.

Statistical Reporting: We have updated Tables 1, 2, and 3 by including F-values and significance levels, enhancing the statistical robustness and transparency of our findings.

Editorial Issues: All highlighted editorial issues have been addressed. For instance, we revised the final paragraph of page 3 to clearly articulate the specific objectives of the study, improving clarity and coherence.

Site Selection and Soil Sampling: We have expanded the discussion of site selection and soil sampling methods by providing detailed descriptions of all land use types, including natural forest land. This addition ensures a more comprehensive understanding of the study context.

We respectfully request that the revised manuscript be considered for publication. We are confident that these revisions address the concerns raised and contribute to improving the quality of the manuscript.

Thank you for your time and valuable input, and we look forward to your decision.

Sincerely,

Abu Regasa (PhD candidate)

Dambi Dollo University

---

## [Editor Report · Decision Letter 2]

4 Dec 2024

Soil Acidity and Fertility Status of Surface Soils under Different Land Uses in Sayo District of Oromia, Western Ethiopia

PONE-D-24-23222R2

Dear Dr. Regasa,

We’re pleased to inform you that your manuscript has been judged scientifically suitable for publication and will be formally accepted for publication once it meets all outstanding technical requirements.

Kind regards,

Dafeng Hui, Ph.D.

Academic Editor

PLOS ONE
---

## [Editor Report · Acceptance letter]

12 Dec 2024

PONE-D-24-23222R2 

PLOS ONE

Dear Dr. Regasa, 

I'm pleased to inform you that your manuscript has been deemed suitable for publication in PLOS ONE. Congratulations! Your manuscript is now being handed over to our production team.

Kind regards, 

on behalf of

Dr. Dafeng Hui 

Academic Editor

PLOS ONE